# Wild-GS: Real-Time Novel View Synthesis from Unconstrained Photo Collections

**Jiacong Xu**
Johns Hopkins University
Baltimore MD 21218, USA
jxu155@jhu.edu

**Yiqun Mei**
Johns Hopkins University
Baltimore MD 21218, USA
ymei7@jhu.edu

**Vishal M. Patel**
Johns Hopkins University
Baltimore MD 21218, USA
vpatel36@jhu.edu

## Abstract

Photographs captured in unstructured tourist environments frequently exhibit variable appearances and transient occlusions, challenging accurate scene reconstruction and inducing artifacts in novel view synthesis. Although prior approaches have integrated the Neural Radiance Field (NeRF) with additional learnable modules to handle the dynamic appearances and eliminate transient objects, their extensive training demands and slow rendering speeds limit practical deployments. Recently, 3D Gaussian Splatting (3DGS) has emerged as a promising alternative to NeRF, offering superior training and inference efficiency along with better rendering quality. This paper presents *Wild-GS*, an innovative adaptation of 3DGS optimized for unconstrained photo collections while preserving its efficiency benefits. *Wild-GS* determines the appearance of each 3D Gaussian by their inherent material attributes, global illumination and camera properties per image, and point-level local variance of reflectance. Unlike previous methods that model reference features in image space, *Wild-GS* explicitly aligns the pixel appearance features to the corresponding local Gaussians by sampling the triplane extracted from the reference image. This novel design effectively transfers the high-frequency detailed appearance of the reference view to 3D space and significantly expedites the training process. Furthermore, 2D visibility maps and depth regularization are leveraged to mitigate the transient effects and constrain the geometry, respectively. Extensive experiments demonstrate that *Wild-GS* achieves state-of-the-art rendering performance and the highest efficiency in both training and inference among all the existing techniques. The code can be accessed via https://github.com/XuJiacong/Wild-GS

## 1 Introduction

With the development of 3D scene representation technologies, novel view synthesis aiming to create photo-realistic images from arbitrary viewpoints is becoming increasingly popular in computer vision. Neural Radiance Field (NeRF) (Mildenhall et al., 2021), as a physically inspired approach, representing the scene by radiance field and density for volume rendering, has accomplished ground-breaking synthesis quality on a range of complex scenes. Many subsequent works extend the applications of NeRF (Haque et al., 2023; Huang et al., 2022; Poole et al., 2022; Yuan et al., 2022) and further improve its robustness (Verbin et al., 2022; Mildenhall et al., 2022; Ma et al., 2022). One central assumption of NeRF and other traditional novel view synthesis methods is that the geometry, material, and lighting conditions in the collected images should remain constant. However, the large number of tourist photos on the internet is usually captured at different times and weathers, or with various camera settings, containing variable appearance and transient occluders. Training NeRF with these *in-the-wild* image collections will result in ghosting and over-smoothing artifacts.

To tackle the aforementioned problem, NeRF-W (Martin-Brualla et al., 2021) introduces image-dependent appearance and transient embeddings to model the appearance variation and transient

38th Conference on Neural Information Processing Systems (NeurIPS 2024).

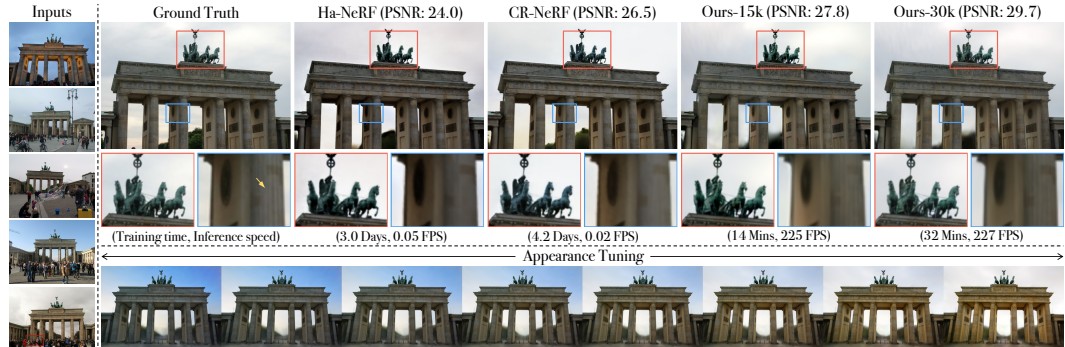

Figure 1: Visual comparison between Wild-GS and other existing approaches (Chen et al., 2022b; Yang et al., 2023). Wild-GS presents superior computational efficiency (tested on single RTX3090), as well as better appearance and geometry reconstruction. Additionally, by modifying the appearance features defined by Wild-GS, one can freely adjust the visual appearance of the entire scene.

uncertainty, which successfully accomplishes high-fidelity and occluder-free rendering. Subsequent advanced variants (Chen et al., 2022b; Yang et al., 2023; Fridovich-Keil et al., 2023; Kassab et al., 2024) have further improved the appearance disentanglement and transfer capability across different views. These approaches decode the appearance variances for all the 3D points with the same latent global vector or directly redistribute the rendered 2D features by global statistics as style transfer (Yang et al., 2023), which cannot explicitly capture the positional-awareness local reflectance. Furthermore, the training cost for implicit representations is extremely high, and the volumetric ray-marching nature of existing methods prevents them from accomplishing real-time rendering.

Although various explicit and hybrid representations have been proposed in recent years to reduce the training cost and expedite the rendering speed of NeRF (Liu et al., 2020; Fridovich-Keil et al., 2022; Yu et al., 2021a; Müller et al., 2022), they usually come with the sacrifice of synthesis quality. Recently, 3D Gaussian Splatting (3DGS) (Kerbl et al., 2023) has revolutionized the realm of novel view synthesis by allowing high-quality and real-time rendering along with competitive training efficiency. Nevertheless, similar to NeRF, the original 3DGS struggles to handle the *in-the-wild* image collections, causing obvious ghosting artifacts and geometry errors. A large amount of Gaussians are placed in the highly occluded areas to model and shade the transient objects, which induces meaningless computations in both the training and rendering stages. Besides, without appearance encoding, 3DGS fails to distinguish the appearance variation between different views. In this paper, we introduce an adaptation of 3DGS, namely Wild-GS, which improves the robustness of 3DGS in dealing with unconstrained images without significant trade-offs on its efficiency merits.

Following the same paradigm as the previous *in-the-wild* approaches, the appearance of each view is decomposed into image-dependent and image-invariant components. Every 3D Gaussian in Wild-GS stores an intrinsic vector to represent the inherent material property around its dominant area, which is invariant to the external environment changes. In addition to the global appearance embedding utilized in prior works, which encodes the universal appearance for all the Gaussians, such as different global illuminations and camera ISP settings, we further align each Gaussian with its corresponding local appearance feature to describe the positional-awareness local variations of reflectance by combining triplane (Chan et al., 2022) and 3DGS representations. Following the nature of 3DGS, the local appearance modeling is implemented in an explicit manner and thus expedites the training process.

The contribution of this paper can be summarized as follows: *i)* We propose a hierarchical appearance decomposition strategy to handle the complicated appearance variances across different views; *ii)* We design an explicit local appearance modeling method to capture the high-frequency appearance details; *iii)* Our model accomplishes the best rendering quality and the highest efficiency in training and inference; *iv)* Our model presents high-quality appearance transfer from arbitrary images.

## 2 Related Work

### 2.1 3D Scene Representation

**Neural Radiance Field.** Synthesizing arbitrary views of a scene from multi-view images is a long-standing research topic in computer vision and graphics. Photo-realistic rendering of novel view

requires accurate reconstruction of 3D geometry and appearance. Early approaches represent the 3D scenes by explicit mesh (Waechter et al., 2014; DEBEC, 1996; Liu et al., 2019) or voxel (Kutulakos & Seitz, 2000; Seitz & Dyer, 1999; Szeliski & Golland, 1998), leading to geometry and appearance inconsistency in complex scenarios (Gao et al., 2022b). Neural Radiance Field (NeRF) (Mildenhall et al., 2021) utilizes an interpolation approach between different views to parse the scene information by neural networks implicitly and provides revolutionary impact. Subsequent attempts (Barron et al., 2021, 2022; Yu et al., 2021b; Xu et al., 2022; Barron et al., 2023) further improve the modeling capability and rendering quality of the original NeRF. To mitigate the resource consumption of large MLP training and inference, advanced radiance field representations such as voxel grid (Liu et al., 2020; Sun et al., 2022a; Fridovich-Keil et al., 2022; Sun et al., 2022b; Hedman et al., 2021), octree (Yu et al., 2021a; Wang et al., 2022; Bai et al., 2023), planes (Chan et al., 2022; Cao & Johnson, 2023; Chen et al., 2022a; Fridovich-Keil et al., 2023) and hash grid (Müller et al., 2022), were investigated. However, the volumetric ray-marching nature of NeRF-based methods involves costly computation due to dense queries along each ray and thus restricts their rendering speed.

**3D Gaussian Splatting**. Recently, 3D Gaussian Splatting (3DGS) (Kerbl et al., 2023) is emerging as a promising alternative to NeRF by presenting impressive efficiency and higher rendering quality (Chen & Wang, 2024). Avoiding unnecessary computation in empty space, 3DGS represents the scene by millions of controllable 3D Gaussians and accomplishes real-time novel view rendering by directly projecting Gaussians within FOV to the 2D screen (Zwicker et al., 2001). Inspired by its efficiency advantages, 3DGS is being employed to replace NeRF in various vision and graphics applications, such as autonomous driving (Zhou et al., 2023b; Yan et al., 2024; Zhou et al., 2024), text-to-3D generation (Chen et al., 2023b; Chung et al., 2023; Liu et al., 2023; Ling et al., 2023), mesh extraction and physical simulation (Guédon & Lepetit, 2023; Xie et al., 2023), controllable 3D scene editing (Chen et al., 2023a; Fang et al., 2023; Zhou et al., 2023a), sparse view reconstruction (Szymanowicz et al., 2023; Charatan et al., 2023; Li et al., 2024), and human avatar (Hu et al., 2023; Shao et al., 2024; Li et al., 2023). This paper embraces the high efficiency of 3DGS and equips it with highly explicit appearance control to achieve fast training and rendering from unconstrained image collections. Notably, improvements in different aspects of the robustness of 3DGS are still under-explored (Zhao et al., 2024; Darmon et al., 2024; Meng et al., 2024).

### 2.2 Novel View Synthesis *in-the-wild*

Traditional novel view synthesis methodologies assume that the geometry, material, and lighting are static in the world, but the *in-the-wild* images collected from internet severely violate this assumption. To resolve this issue, NRW (Meshry et al., 2019) applies a rerendering network to merge the semantic mask and latent appearance embedding. Differently, NeRF-W (Martin-Brualla et al., 2021) leverages the implicit radiance field to represent the 3D scene and attaches transient and appearance embeddings for each training image to handle the environmental variations. Instead of optimizing the appearance for each inference image, Ha-NeRF (Chen et al., 2022b) encodes the images into latent appearance features by a CNN and employs a 2D visibility map conditioning on a learnable transient vector to emphasize static objects. Rendering the color of a pixel by a single ray may lose the global information across multiple pixels. CR-NeRF (Yang et al., 2023) utilizes grid sampling to render cross-ray features and transforms them by the global statistics derived from the reference view. More recently, $K$-planes (Fridovich-Keil et al., 2023), an extension of triplane (Chan et al., 2022), is implemented to factorize the variable radiance field into 2D planes and an appearance vector. After fitting the $K$-planes to the scene, RefinedFields (Kassab et al., 2024) additionally introduces a scene refining stage to refine the plane features by Stable Diffusion (Rombach et al., 2022). The aforementioned methods relying on implicit representations require costly training, and their rendering speed is also limited by the ray-marching nature of the radiance field.

## 3 Preliminaries

### 3.1 3D Gaussian Splatting

A collection of trainable 3D Gaussians is leveraged by 3D Gaussian Splatting (3DGS) (Kerbl et al., 2023) to represent the scene explicitly. The position and shape of each 3D Gaussian are controlled by its center $\mu \in \mathbb{R}^3$ and a 3D covariance matrix $\Sigma \in \mathbb{R}^{3\times 3}$ in world coordinates, respectively. $\Sigma$ is decomposed into scaling $s$ and rotation $r$ to preserve positive semi-definite property. These Gaussains

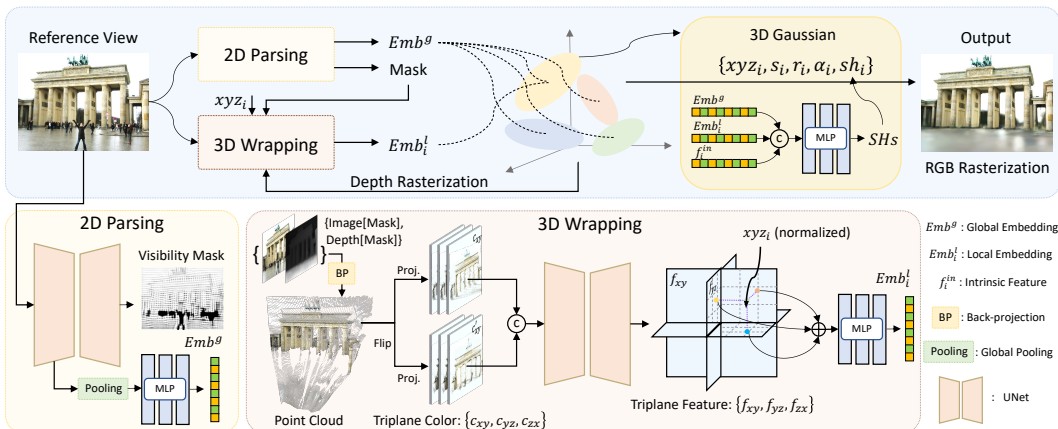

Figure 2: Overview of the architecture of our proposed Wild-GS. The reference view is first processed by a 2D Parsing module to extract the visibility mask and global appearance embedding. Given the mask and rendered depth from 3DGS, we back-project the 2D reference image without transient objects to the space and construct the static 3D point cloud. Then, these 3D points are re-projected to three predefined orthogonal planes using their normalized coordinates for generation of triplane features. Each 3D Gaussian queries its local appearance embedding by providing the spatial coordinate to the 3D Wrapping module. With the global and local embeddings and the stored intrinsic feature, we can predict the SH coefficients $sh$ of every 3D Gaussian for RGB rasterization.

are directly projected to the screen for high-speed rendering, called Splatting (Zwicker et al., 2001). Given the world-to-camera transformation matrix $\mathbf{W}$ and Jacobian of the affine approximation of perspective transformation $\mathbf{J}$, the projected 2D covariance matrix can be computed by:

$$\Sigma' = \mathbf{J}\mathbf{W}\Sigma\mathbf{W}^\top\mathbf{J}^\top. \tag{1}$$

Each Gaussian stores a learnable opacity $\alpha_i$ and a set of spherical harmonic (SH) coefficients for view-dependent color $c_i$ computation. After sorting the Gaussians by depth, alpha compositing is utilized to compute the final color for each pixel, which can be written as:

$$C = \sum_{i=1}^{n} c_i \alpha_i' \prod_{j=1}^{i-1}(1 - \alpha_j'), \tag{2}$$

where $\alpha_i'$ is the multiplication of $\alpha_i$ and splatted 2D Gaussian. Heuristic point densification and pruning are employed in the training process for efficient 3D scene representation.

### 3.2 Triplane Representation

The hybrid explicit-implicit triplane representation was first introduced by EG3D (Chan et al., 2022) to enable 2D GAN (Goodfellow et al., 2020) to possess the capability of 3D generation. Many subsequent works (Gao et al., 2022a; Wang et al., 2023; Shue et al., 2023) further demonstrate the superiority of triplane on text-to-3D or 2D-to-3D generation tasks. A triplane consists of three individual 2D feature maps representing three orthogonal planes $P_{XY}$, $P_{YZ}$, and $P_{ZX}$ in 3D space. The feature of a 3D point $p$ can be queried by projecting the point onto three planes and summing up the interpolated individual plane features $f_p = f_{xy} + f_{yz} + f_{zx}$. In the original implementation, an MLP is attached to decode the feature into density and color for volume rendering.

## 4 Wild-GS

The objective of Wild-GS is to improve the robustness of 3DGS in handling *in-the-wild* photo collections without loosing much on its efficiency benefits. Specifically, Wild-GS applies a heuristic decomposition on the appearances of space points to accomplish hierarchical and explicit appearance control for each 3D Gaussian. Figure 2 illustrates the pipeline of Wild-GS, which mainly consists of three components: *i)* 2D Parsing Module extracts the high-level 2D appearance information and predicts the mask for static objects; *ii)* 3D Wrapping Module constructs the positional-awareness local

appearance embedding for each 3D Gaussian; *iii)* A fusion network is shared for all the Gaussians merges and decodes the appearance features for adaptive SHs prediction.

## 4.1 Hierarchical Appearance Modeling

In this section, we propose a hierarchical appearance modeling framework for 3DGS that adaptively generates specific appearance embedding for individual 3D Gaussian. In this framework, the appearance of each Gaussian for a given reference view is determined by three components: *(a)* Global appearance embedding $Emb^g$ capturing the illumination level or tone mapping of the entire scene; *(b)* Local appearance embedding $Emb^l_i$ describing the positional-aware local reflectance for $i$-th 3D Gaussian; *(c)* Intrinsic feature $f^{in}_i$ storing the inherent attributes of the material in the dominant area for each Gaussian. Before rasterization, a shared fusion network $M^F_\theta$ is leveraged to decode the view-dependent color $sh$ from these three appearance components:

$$sh_i = M^F_\theta(Emb^g \oplus Emb^l_i \oplus f^{in}_i), \tag{3}$$

where $\oplus$ refers to concatenate operation. This heuristic appearance decomposition allows efficient and effective appearance control from the entire scene to local areas, incorporating commonalities and specificities of different Gaussians. Without modifying the rasterization process, the superior inference efficiency of the original 3DGS is preserved after caching the $sh$ for every Gaussian.

### 4.1.1 Global Appearance Encoding

Tourist photos collected on the internet are usually captured in different weathers and times or with various cameras, resulting in obvious appearance variations in the photo collections. Most of these variations are shared by different areas of the scene and determined by common environmental factors, e.g., the lighting level at the time of shooting. Furthermore, different post-processing processes of the photograph devices, such as gamma correction, exposure adjustment, and tone mapping, lead to various visual effects in the captured scene. These external and internal factors globally influence the appearance and are invariant to the positions of the space points. Therefore, we apply the global appearance embedding $Emb^g$ extracted from the given reference image $I_R$ to all the 3D Gaussians to model the low-frequency appearance changes among the entire scene. This encoding process is implemented by directing the feature maps $F_{I_R}$ obtained from the UNet encoder in the 2D Parsing module through a global average pooling layer, followed by a trainable MLP $M^G_\theta$:

$$Emb^g = M^G_\theta(AvgPooling(F_{I_R})). \tag{4}$$

### 4.1.2 Local Appearance Control

The physical interaction between 3D scenes and their environments presents long-standing challenges in computer graphics. For instance, varying directions of light can create distinct specular highlights and shadows in specific areas. The global appearance embedding is inadequate to model the detailed and high-frequency local appearance changes, especially for 3DGS with explicit and discrete representation. Thus, we design a local appearance control strategy to explicitly align the appearance clues from the reference image to the corresponding 3D Gaussians by combining the triplane and 3DGS representations. Specifically, each 3D Gaussian can query its local appearance embedding $Emb^l_i$ from the generated triplane feature maps using its learned center position $\mu_i = xyz_i$.

**Triplane Color Creation.** Unlike previous works (Wang et al., 2023; Zou et al., 2023) that learn triplane features in generative ways, Wild-GS leverages the color information from the reference view to infer high-dimensional triplane maps. To capture the 3D local appearance, we first back-project the reference image into the space using the rendered depth $\hat{D}_{I_R}$ and camera parameters $\omega_{I_R}$. Here, the visibility mask $M_{I_R}$ is required to exclude transient objects. This step can be implemented as:

$$\hat{D}_{I_R} = \sum_{i=1}^{n} d_i \alpha'_i \prod_{j=1}^{i-1}(1 - \alpha'_j), \tag{5}$$

$$\{C_{I_R}, P_{I_R}\} = BP(I_R[M_{I_R} > Th], \hat{D}_{I_R}[M_{I_R} > Th], \omega_{I_R}), \tag{6}$$

where $Th$ defines the threshold distinguishing the transient and static objects in the visibility mask and $\{C_{I_R}, P_{I_R}\}$ refers to the generated point cloud with positions $P_{I_R}$ and colors $C_{I_R}$. Then, the 3D points are normalized for re-projection onto the three orthogonal planes defined by the triplane:

$$\{c_{xy}, c_{yz}, c_{zx}\} = Proj(C_{I_R}, \widetilde{P}_{I_R}), \tag{7}$$

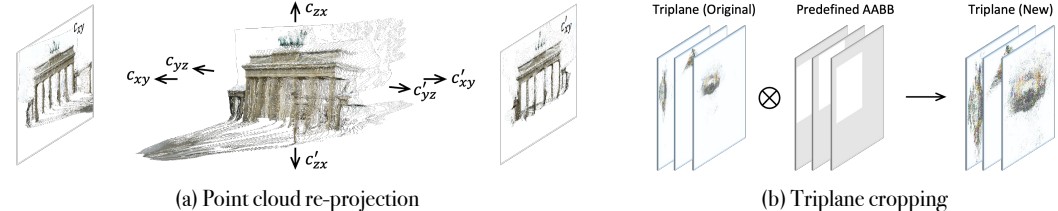

(a) Point cloud re-projection        (b) Triplane cropping

Figure 3: (a) The point cloud from the reference image is projected along three axes and their reverses to generate the triplane color; (b) Illustration of the distribution of the 3D Gaussians on the original triplane and cropped one. Axis-aligned bounding box (AABB) is utilized to accomplish 3D cropping.

where $\widetilde{P}_{I_R}$ represents the normalized point positions. Viewing an object from one side can result in the loss of information from the opposite side. While three orthographic views capture most of the scene's geometric information, we've empirically observed that a simple network often struggles to learn the complex multi-view correlations. Therefore, we concatenate the projections along each axis and their corresponding reverses, denoted by $\{c'_{xy}, c'_{yz}, c'_{zx}\}$, to form the triplane color (Figure 3-(a)).

The triplane color is further processed by a UNet $U_\theta^{3D}$ to extract the triplane feature maps $F_{I_R}^T$, which then are utilized to interpolate the three plane features $\{f_{xy}^i, f_{yz}^i, f_{zx}^i\}$ for each 3D Gaussian.

$$F_{I_R}^T = U_\theta^{3D}(\{c_{xy}, c_{yz}, c_{zx}\} \oplus \{c'_{xy}, c'_{yz}, c'_{zx}\}). \tag{8}$$

The positional-awareness local appearance embedding can be obtained by feeding the summation of three plane features to an MLP $M_\theta^L$:

$$Emb_i^l = M_\theta^L(f_{xy}^i + f_{yz}^i + f_{zx}^i). \tag{9}$$

**Efficient Triplane Sampling.** The resolution of the triplane feature maps must be sufficiently high to accurately describe the detailed variations in local appearance. However, sampling from high-resolution maps is computationally expensive in terms of time and space. As depicted in Figure 3-(b), the Gaussian points projected onto the triplane maps tend to concentrate within a small area. To reduce the sampling cost and improve the utilization of the triplane maps, we apply an axis-aligned bounding box (AABB) to confine the 3D space containing most of the 3D Gaussians. As a compliment, we set $Emb_i^l = v$, where $v$ is a learnable vector, for Gaussians outside the predefined AABB.

### 4.1.3 Learnable Intrinsic Feature

Beyond image-based appearance modeling strategies, Wild-GS maintains a learnable intrinsic appearance feature for each 3D Gaussian. This feature characterizes the inherent material properties within its dominant area, which remain consistent despite environmental changes. This approach is inspired by EAGLES (Girish et al., 2023), which compresses the attributes of the 3D Gaussian into a low-dimensional latent vector. By separating internal and external appearances, this heuristic decomposition effectively enhances the rendering quality of Wild-GS in our experiments.

### 4.2 Depth Regularization

Depth information of the training images has been widely employed in the sparse view reconstruction methods (Deng et al., 2022; Li et al., 2024; Zhu et al., 2023). Different from NeRF, 3DGS with unstructured representation is sensitive to geometric regularization. In Wild-GS, the rendered depth is leveraged to back-project the reference view, so it determines the precision of the generated point cloud. Thus, we also incorporate the depth regularization strategy to constrain the geometry of the scene. Specifically, Depth Anything (Yang et al., 2024) is employed here to estimate the monocular depth $D_{I_R}^{Est}$ for each reference view. Besides, we modify the Pearson correlation loss proposed by FSGS (Zhu et al., 2023) by masking out the transient objects and applying it in Wild-GS:

$$\mathcal{L}^D = \frac{Cov(\hat{D}_{I_R}[M_{I_R} > Th], D_{I_R}^{Est}[M_{I_R} > Th])}{\sqrt{Var(\hat{D}_{I_R}[M_{I_R} > Th]) \cdot Var(D_{I_R}^{Est}[M_{I_R} > Th])}}. \tag{10}$$

| Method | Brandenburg Gate | | | Sacre Coeur | | | Trevi Fountain | | | Efficiency | |
|---|---|---|---|---|---|---|---|---|---|---|---|
| | PSNR | SSIM | LPIPS | PSNR | SSIM | LPIPS | PSNR | SSIM | LPIPS | Training | Inference |
| 3DGS | 19.63 | 0.8817 | 0.1378 | 17.95 | 0.8455 | 0.1633 | 17.23 | 0.6963 | 0.2815 | **0.12** | 220 |
| NeRF-W | 24.17 | 0.8905 | 0.1670 | 19.20 | 0.8076 | 0.1915 | 18.97 | 0.6984 | 0.2652 | 60.3 | 0.05 |
| Ha-NeRF | 24.04 | 0.8873 | 0.1391 | 20.02 | 0.8012 | 0.1710 | 20.18 | 0.6908 | 0.2225 | 71.6 | 0.05 |
| CR-NeRF | 26.53 | 0.9003 | 0.1060 | 22.07 | 0.8233 | 0.1520 | 21.48 | 0.7117 | 0.2069 | 101 | 0.02 |
| Wild-GS$^\dagger$ | 27.81 | 0.9180 | 0.1085 | 23.85 | 0.8567 | 0.1575 | 22.77 | 0.7629 | 0.2229 | 0.23 | 225 |
| Wild-GS | **29.65** | **0.9333** | **0.0951** | **24.99** | **0.8776** | **0.1270** | **24.45** | **0.8081** | **0.1622** | 0.52 | **227** |
| w/o Crop | 29.15 | 0.9324 | 0.0968 | 24.93 | 0.8761 | 0.1323 | 23.61 | 0.7915 | 0.1856 | 0.85 | 223 |
| w/o $\{c'\}$ | 28.52 | 0.9286 | 0.1019 | 24.37 | 0.8623 | 0.1379 | 24.14 | 0.8036 | 0.1657 | 0.47 | 227 |
| w/o Global | 29.00 | 0.9295 | 0.1006 | 24.82 | 0.8761 | 0.1309 | 24.18 | 0.8045 | 0.1702 | 0.53 | 224 |
| w/o Mask | 28.85 | 0.9292 | 0.0981 | 24.71 | 0.8833 | 0.1216 | 23.91 | 0.8063 | 0.1590 | 0.53 | 205 |
| w/o Depth | 28.36 | 0.9261 | 0.1064 | 23.36 | 0.8573 | 0.1672 | 23.20 | 0.7878 | 0.1827 | 0.48 | 218 |

Table 1: Quantitative experimental results of existing methods (NeRF-W (Martin-Brualla et al., 2021), Ha-NeRF (Chen et al., 2022b), and CR-NeRF (Yang et al., 2023)) and Wild-GS on Phototourism dataset. Wild-GS$^\dagger$ indicates that the model is trained by 15k iterations. The efficiencies of different methods are quantified by their training times (hours) and inference speeds (frame per second). Crop, Global, Mask, and Depth are abbreviations for triplane cropping, global appearance encoding, transient mask prediction, and depth regularization, respectively. $\{c'\}$ refers to $\{c'_{xy}, c'_{yz}, c'_{zx}\}$.

### 4.3 Handling Transient Objects

The view-inconsistent transient objects widely exist in unconstrained image collections, requiring the model to put more useless and even malicious efforts into representing their appearance, especially for 3DGS, where the geometry and the appearance are highly entangled. Similar to CR-NeRF (Yang et al., 2023), we also leverage a visibility mask to indicate the easier exemplars, the static objects, by feeding the reference view to the UNet $U_\theta^{2D}$ in the 2D Parsing module and adaptively predicting the visibility mask $M_{I_R}$. The training of $U_\theta^{2D}$ is implemented in an unsupervised manner by forcing the rendering loss to focus only on the static objects. Additional mask regularization is utilized to prevent meaningful pixels from being masked. The detailed implementation can be written as:

$$\mathcal{L}^I = \lambda^I |I_R \odot M_{I_R} - \hat{I}_R \odot M_{I_R}| + (1 - \lambda^I) \cdot SSIM(I_R \odot M_{I_R}, \hat{I}_R \odot M_{I_R}). \quad (11)$$

$$\mathcal{L}^M = (1 - M_{I_R})^2. \quad (12)$$

For our explicit appearance control strategy, the accuracy of the visibility mask is crucial since the model may project the appearance of the transient objects to the static ones or, conversely, partially mask the appearance of the static objects. Thus, we provide a mask threshold to control the trade-off.

### 4.4 Training Objective

Incorporating all the aforementioned techniques, we can build the training objective for Wild-GS:

$$\mathcal{L}_{total} = \mathcal{L}^I + \lambda^M \mathcal{L}^M + \lambda^D \mathcal{L}^D. \quad (13)$$

Notably, in the initial training stage, the prediction of $M_{I_R}$ is not sufficiently accurate. Therefore, to expedite the training speed and remedy the transient effect, the depth regularization and explicit appearance control strategies, whose functionalities are highly dependent on the mask, are not used.

## 5 Experimental Results

**Implementation, Datasets, and Evaluation.** We develop our method based on the original implementation of 3DGS (Kerbl et al., 2023). All the networks in Wild-GS are optimized by Adam optimizer (Kingma & Ba, 2014). The hyper-parameter $\lambda^M$ is reduced linearly to effectively remove the transient objects and stabilize the training process. Following previous works (Chen et al., 2022b; Yang et al., 2023), we evaluate different methods on three *in-the-wild* datasets: "Brandenburg Gate", "Sacre Coeur", and "Trevi Fountain" extracted from the Phototourism dataset and downsample the images by 2 times ($R/2$). All the training times and inference speeds are tested on a single RTX3090 for fair comparison. In addition to comparing existing approaches, we also provide ablation studies for different model components to validate their effectiveness.

### 5.1 Comparison Experiments

**Quantitative Comparison.** Original 3DGS is not equipped with appearance modeling modules, resulting in lower rendering performance on *in-the-wild* datasets. As shown in Table 1, our adaptation

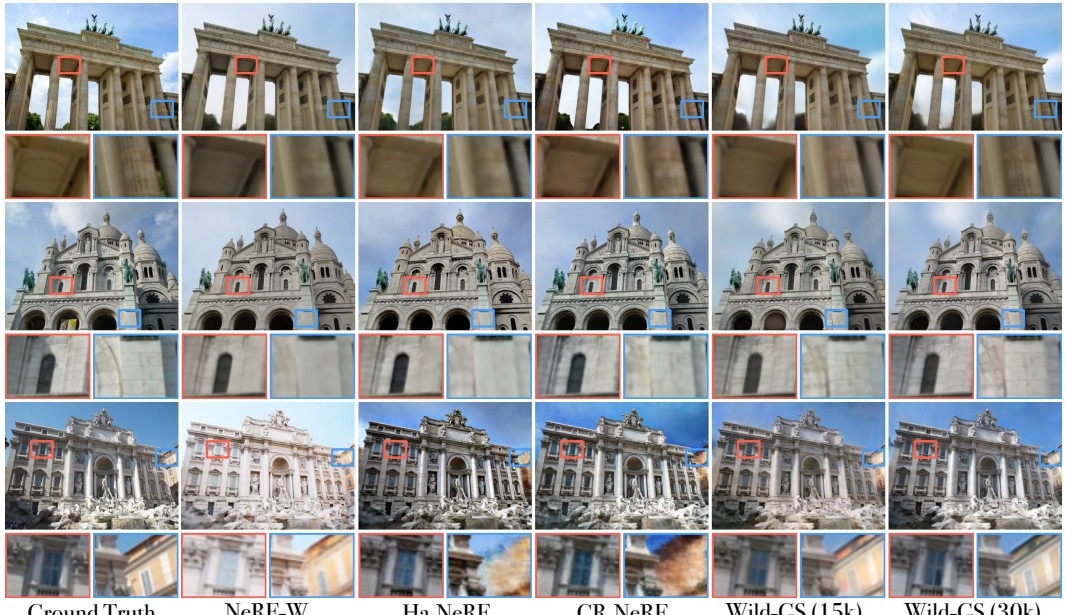

| Ground Truth | NeRF-W | Ha-NeRF | CR-NeRF | Wild-GS (15k) | Wild-GS (30k) |

Figure 4: Visual comparison of rendering quality between different approaches. Red and blue crops mainly emphasize appearance and geometry differences, respectively.

Wild-GS successfully improves the capability and robustness of 3DGS and accomplishes superior rendering quality and efficiency compared with existing approaches. Specifically, Wild-GS presents around 3 PSNR increase along with $200\times$ shorter training time and $10000\times$ faster rendering speed versus the previous state-of-the-art model CR-NeRF. Furthermore, Wild-GS has achieved the highest PSNR and SSIM among all the methods with only 14 minutes of training (15k iterations).

**Qualitative Comparison.** The advanced representation of 3DGS has the potential to recover the high-frequency appearance and geometry details of the scene. Built upon 3DGS, our method Wild-GS also shows more accurate reconstructions of local appearance and geometry along with better global appearance modeling capability compared with other NeRF-based methods (Figure 4).

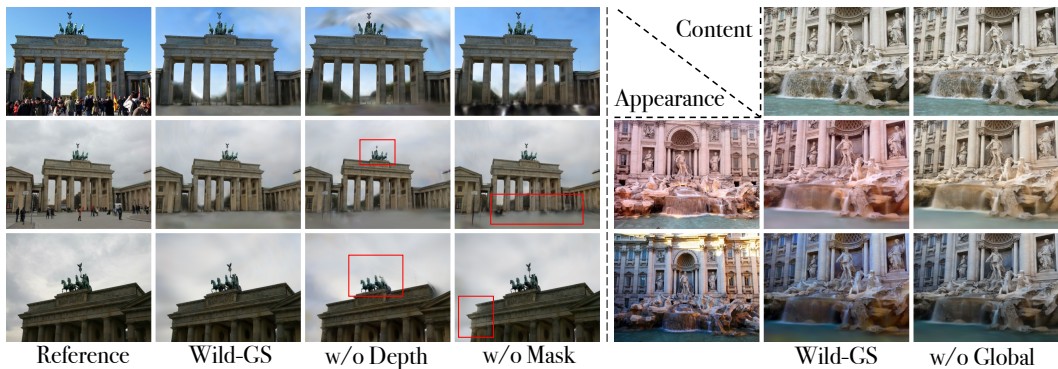

| Reference | Wild-GS | w/o Depth | w/o Mask | | Wild-GS | w/o Global |

Figure 5: Rendering results of ablation study on Wild-GS when removing depth regularization, transient mask (left), and global appearance encoding (right). Red rectangles indicate the areas where geometry is missing or color inconsistency happens. Notations follow Table 1
.

## 5.2 Ablation Study

To explicitly model the complicated appearance variances of unconstrained photos and follow the nature of 3DGS, we leverage triplane to generate the position-awareness local appearance feature for each 3D Gaussian. Additional re-projections along opposite directions $\{c'_{xy}, c'_{yz}, c'_{zx}\}$ are combined with original $\{c_{xy}, c_{yz}, c_{zx}\}$ to constitute the color triplane. Table 1 indicates that this operation effectively improves the rendering quality with slight trade-offs in training efficiency. Besides, by

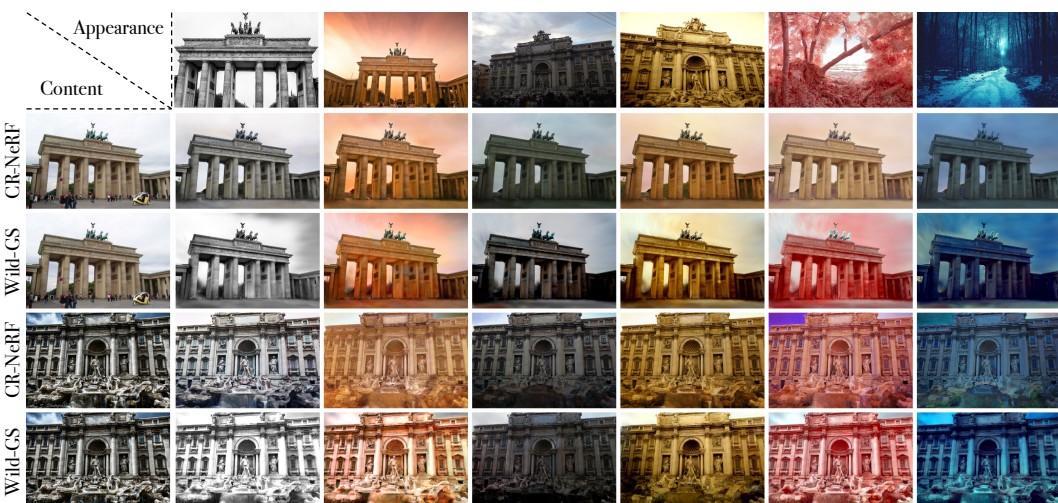

Figure 6: Appearance and style transfer to novel views using reference images inside and outside the training dataset. Two arbitrary style images are borrowed from Ha-NeRF (Chen et al., 2022b).

cropping the triplane and confining the sampling, Wild-GS significantly reduces the training time by around 39% while preserving and even improving the rendering quality.

Depth regularization is utilized in the training process to constrain the geometry and stabilize our explicit appearance modeling strategy. Wild-GS cannot align appearance features to corresponding Gaussians without accurate depth information, causing performance degradation and missing geometry in the final reconstruction (Figure 5). Without the transient mask prediction, ghosting effects and color inconsistencies are observed in the areas highly occluded by transient objects.

In terms of the metrics, leveraging global appearance modeling cannot obtain obvious improvements in rendering quality. The underlying reason is that the re-projected color triplane already contained all the appearance information for the reference viewpoint. However, as shown in Figure 5, Wild-GS (w/o Global) fails to capture the global appearance statistics and struggles to transfer the global color tone to another view. Thus, both embeddings are critical to the robustness of Wild-GS.

## 5.3 Appearance Transfer

In addition to the reference-based view synthesis task, Ha-NeRF and CR-NeRF extended the application of *in-the-wild* methods to appearance transfer and even style transfer of 3D scenes, which further validates their appearance modeling capabilities. Figure 6 contains the qualitative comparison of Wild-GS and CR-NeRF on this new task. For most appearance (style) images, Wild-GS can successfully capture the overall color tone and transfer it to novel views. Compared with CR-NeRF, our method accomplishes more accurate and robust appearance modeling and presents more color-consistent renderings. Furthermore, by linearly combining the appearance features extracted from two different reference views, one can freely tune the appearance of the 3D scene (Figure 1).

## 6 Conclusion

In this paper, we introduce *Wild-GS*, which adapts 3DGS to handle unconstrained photo collections without significant trade-offs on its efficiency benefits. Specifically, Wild-GS hierarchically decomposes the appearance of a given reference view into image-based global and local appearance embeddings and image-invariant intrinsic appearance features for each Gaussian. Following the nature of 3DGS, we leverage triplane representation to accomplish explicit local appearance modeling and allow Gaussians to sample their triplane features according to their specific positions. Triplane generation and sampling modifications are proposed to improve the rendering quality and training efficiency. Depth regularization and transient object handling are employed for better geometry and color consistency. Extensive experiments demonstrate that *Wild-GS* achieves state-of-the-art rendering performance and the highest efficiency on training and inference among all the existing *in-the-wild* techniques. Besides, applications for appearance transfer and tuning are provided.

# 7 Acknowledgement

This research is based upon work supported by the Office of the Director of National Intelligence (ODNI), Intelligence Advanced Research Projects Activity (IARPA), via IARPA R&D Contract No. 140D0423C0076. The views and conclusions contained herein are those of the authors and should not be interpreted as necessarily representing the official policies or endorsements, either expressed or implied, of the ODNI, IARPA, or the U.S. Government. The U.S. Government is authorized to reproduce and distribute reprints for Governmental purposes notwithstanding any copyright annotation thereon.

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

# A    Comparison with Concurrent *in-the-wild* 3DGS

There are two concurrent works, GS-W (Zhang et al., 2024) and SWAG (Dahmani et al., 2024), focusing on the adaptation of 3DGS to the *in-the-wild* setting at the time when we submit this paper. While GS-W leverages adaptive sampling on 2D feature maps to get dynamic appearance embedding for each Gaussian, their methods are still constrained in the 2D space without fully explicit appearance control. In terms of rendering performance, our method surpasses GS-W by around $1.5$ PSNR and SWAG by 2 PSNR on Phototourism datasets. Besides, GS-W requires 2 hours for training on a single RTX3090 (stated in their paper), while our method only takes around half an hour (32 mins).

| Method | Palace of Westminster | | | Pantheon Exterior | | | Buckingham Palace | | |
|---|---|---|---|---|---|---|---|---|---|
| | PSNR | SSIM | LPIPS | PSNR | SSIM | LPIPS | PSNR | SSIM | LPIPS |
| 3DGS-AE | 21.0781 | 0.8189 | 0.2289 | 22.4432 | 0.8389 | 0.1681 | 23.1788 | 0.8489 | 0.2141 |
| GS-W | 24.7343 | 0.8615 | 0.1782 | 26.1668 | 0.8888 | 0.1022 | 25.6356 | 0.8634 | 0.1671 |
| Wild-GS | **25.8281** | **0.8677** | **0.1635** | **26.7969** | **0.8888** | **0.1018** | **26.7160** | **0.8799** | **0.1592** |
| w/o Local | 21.8409 | 0.8093 | 0.2164 | 21.7973 | 0.8094 | 0.1568 | 23.6144 | 0.8606 | 0.1860 |
| w/o $f^{in}$ | 22.2028 | 0.8014 | 0.2648 | 21.1427 | 0.7751 | 0.2531 | 24.9937 | 0.8500 | 0.2426 |

Table 2: Quantitative experimental results on three extra datasets. 3DGS-AE replaces the appearance encoding of Wild-GS (ours) with a learnable embedding as NeRF-W and optimizes it in an autoencoder way. Local and $f^{in}$ refer to the triplane local appearance embedding and the intrinsic feature.

Besides the three datasets used in the main paper, we also extracted three subsets: "Palace of Westminster", "Pantheon Exterior", and "Buckingham Palace" from Phototourism dataset and implemented more experiments to further demonstrate the strengths of Wild-GS. As shown in Figure 7, Wild-GS provides more accurate appearance modeling compared with GS-W Zhang et al. (2024).

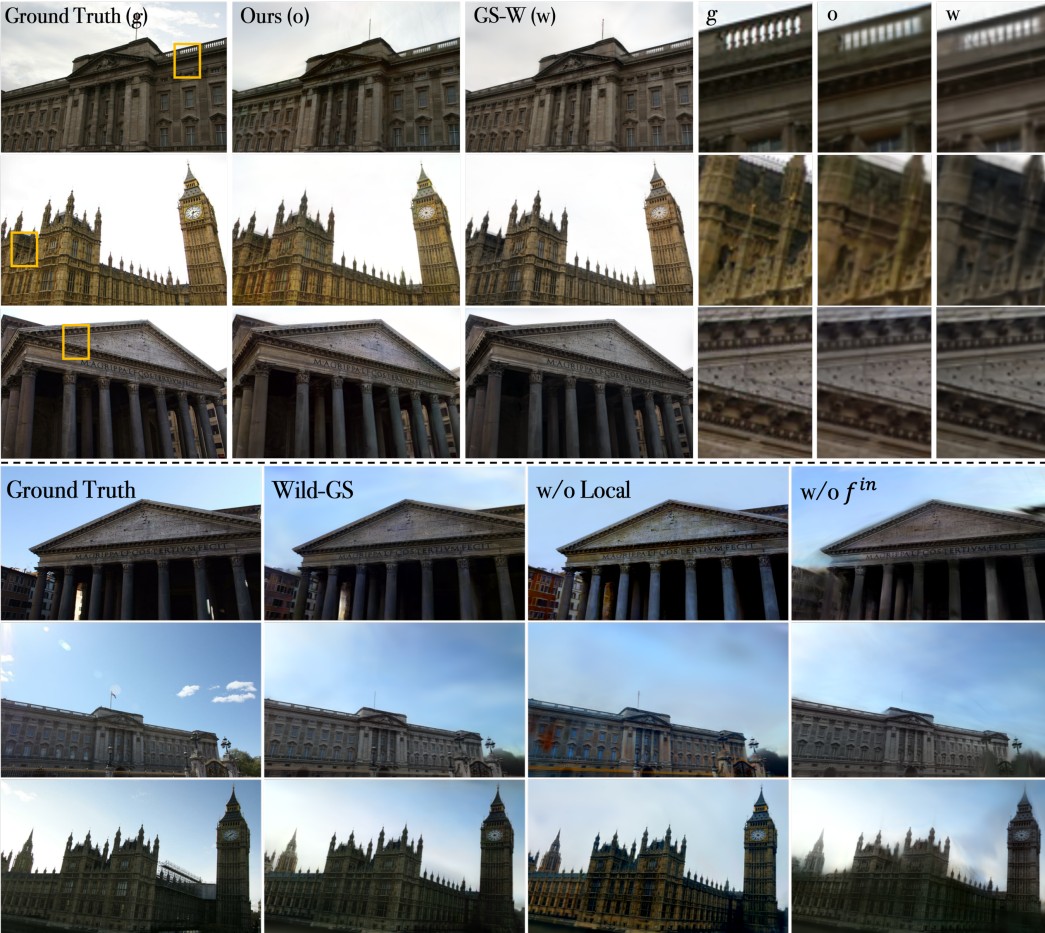

Figure 7: Visual comparison of Wild-GS (ours) and GS-W (concurrent work) and ablation study.

The quantitative results in Table 2 show that Wild-GS archives around 1 PSNR increase compared with GS-W and the local appearance modeling and intrinsic feature inside each Gaussian are required for its superior performance.

## B    More Implementation Details

For semantically meaningful transient mask prediction, we utilize the ResNet-18 pre-trained by ImageNet as the encoder of the UNet in the 2D Parsing module. The ratio of triplane cropping is simply set to $0.5$, and fine-tuning this parameter can achieve a better trade-off in efficiency and rendering quality. The dimensions for global & local appearance embeddings and intrinsic features are set to be $16$ and $32$, respectively. For a fair comparison with 3DGS, we only optimize the hyper-parameters in our attached framework and do not introduce any modifications to their original setting. $\lambda^M$ is linearly reduced from $0.4$ to $0.1$ to stabilize the training process, while $\lambda^D$ is kept constant $0.05$ during the entire training process. Figure 8 demonstrates the transient object handling capability of Wild-GS.

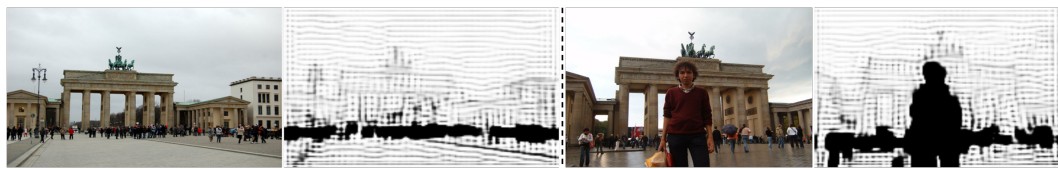

Figure 8: Visualization of the transient masks (learned in an unsupervised way) predicted by Wild-GS.

## C    Detailed Appearance Control

As shown in Figure 9, our method Wild-Gs can capture the high-frequency local appearance details and accomplishes a more accurate local appearance modeling than CR-NeRF, which further demonstrates the effectiveness of our proposed explicit local appearance control strategy.

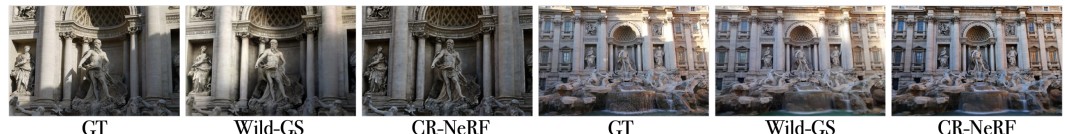

| GT | Wild-GS | CR-NeRF | GT | Wild-GS | CR-NeRF |

Figure 9: Comparison of Wild-GS and CR-NeRF on Local appearance modeling.

## D    Limitation

Similar to previous approaches, Wild-GS still cannot recover the detailed geometry and appearance of the ground (road or sidewalk), causing blur and useless computations in corresponding areas. Besides, since the transient masks are learned in an unsupervised manner, they tend to mask the areas with unusual appearance (hard to model but easy to mask out) and lead to color inconsistency in these areas. We suggest using more advanced segmentation networks to obtain accurate transient masks, which is beneficial to appearance modeling and geometry reconstruction. Even though Wild-GS has achieved high efficiency in training and rendering, it still requires at least double the training time of the original 3DGS to achieve comparable rendering performance. Therefore, further reducing training time without sacrificing rendering quality is a potential and meaningful work in the future.

