# OpenReview forum: "Wild-GS: Real-Time Novel View Synthesis from Unconstrained Photo Collections"
_NeurIPS.cc/2024/Conference — NeurIPS 2024 poster_

### Official Review · Reviewer_E9p7 · 2024-06-13

**Soundness:** 3
**Presentation:** 3
**Contribution:** 2
**Rating:** 3
**Confidence:** 5

**Summary:**

Wild-GS proposes a heuristic appearance decomposition strategy to deal with arbitrary images captured in the wild. Specifically, the authors decompose the appearance of each Gaussian into three components: global appearance, local appearance, and intrinsic features. Compared to existing methods, this paper achieves the highest visual quality and the fastest training and rendering speed.

**Strengths:**

1. the paper is well-written and easy to understand.
2. this paper proposes a novel hierarchical appearance decomposition method, achieving high-quality appearance transfer from arbitrary images (no matter inside and outside the training dataset).
3. Wild-GS achieves SOTA performance on three in-the-wild datasets compared with baselines (NeRF-W, Ha-NeRF, CR-NeRF)

**Weaknesses:**

1. **Lack of Novelty**. First, this task is boring because it has already been successfully addressed in NeRF, making it likely that it can also be applied to 3D-GS. Second, in line 67, the authors summarize the contributions into four points. However, I believe that the first, second, and fourth contributions can be considered as a single contribution, while the third contributions are inherited from 3D-GS. Therefore, this paper can be seen as having only one main contribution. Third, although the authors discuss the comparison with concurrent in-the-wild 3D-GS works, they do not directly compare with them due to the absence of released code. However, Scaffold-GS [1] and Octree-GS [2] can also handle in-the-wild images effectively due to their appearance MLP. Additionally, VastGaussian [3] also proposes an appearance embedding module by the CNN network. Considering this, I believe that Wild-GS may not surpass them in terms of visual quality if not consider the transient objects.

2. **Confused of Module Design**. I struggle to understand the motivation for using a triplane to represent local appearance. In line 14, the authors clarify that they aim to explicitly align pixel appearance features with corresponding local Gaussians. This raises a question: it appears that the authors require only a continuous volume representation. Therefore, any representation that provides a continuous volume, such as triplane, vector and plane components like TensoRF [4] or hash-encoding, could be suitable.

3. **Some sentences seem to overstate their claims.** In lines 160 and 182, the authors describe a local appearance design intended for physical interactions, such as distinct specular highlights and shadows. However, I have not seen any experiments that demonstrate this, and there is even no ablation experiment without the local feature. Similarly, in lines 161 and 216, the authors explain that they maintain a learnable intrinsic feature, which is said to represent inherent material properties. I am curious to know what the results would be if there were no global and local appearance features.


[1] Scaffold-gs: Structured 3d gaussians for view-adaptive rendering

[2] Octree-gs: Towards consistent real-time rendering with lod-structured 3d gaussians

[3] VastGaussian: Vast 3D Gaussians for Large Scene Reconstruction

[4] Tensorf: Tensorial radiance fields

**Questions:**

1. I am curious to know about the visual quality, training speed, and rendering speed compared to Scaffold-GS.
2. If the triplane representation is replaced with hash encoding, how would the performance be affected?
3. It would be beneficial if the authors provided an ablation experiment without the local feature, as it can demonstrate the effectiveness of the local appearance design in modeling distinct specular highlights and shadows.
4. I am interested in knowing the results if there were no global and local appearance features, as it would provide insights into the quality of the decomposition.

**Limitations:**

the authors discuss the limitation in the appendix.

---

> ### Author Rebuttal · Authors · 2024-08-07
>
> We sincerely appreciate your thoughtful comments. Below, we address your questions and concerns. Similar questions and weaknesses are merged.
>
> ---
> **W**: " This task is boring because it has already been successfully addressed in NeRF, making it likely that it can also be applied to 3D-GS."
>
> **A**: Even though several existing baselines based on NeRF have improved the performance for handling in-the-wild photo collections, they all show common issues in appearance modeling and training or inference efficiency. Simply replacing NeRF with 3DGS cannot give good results, considering the former designs are mainly suitable for implicit representation. Therefore, it is worth studying how to reasonably adapt 3DGS to handle in-the-wild photos following the nature of 3DGS without losing too much of its efficiency. Our experimental results show that our model surpasses previous SOTA by a big margin and significantly improves the capacity of 3DGS for handling in-the-wild images.
>
> ---
> **W**: "Although the authors discuss the comparison with concurrent in-the-wild 3D-GS works, they do not directly compare with them due to the absence of released code."
>
> **A**: During the review process, one of the concurrent work (GS-W [1]) released its code. Thus, we provide the comparison results in the pdf of our rebuttal, please check it. Our model still performs better than other methods.
>
> [1] Gaussian in the Wild: 3D Gaussian Splatting for Unconstrained Image Collections
>
> ---
> **W**: "Scaffold-GS and Octree-GS can also handle in-the-wild images effectively due to their appearance MLP. Additionally, VastGaussian also proposes an appearance embedding module by the CNN network."
>
> **A**: Our design in Wild-GS is mainly from the external, so one can change the original 3DGS to other advanced 3DGS models, such as Scaffold-GS and Octree-GS, freely. We will provide the results by replacing the 3DGS with Scaffold-GS in the Appendix for the reader's benefit. VastGaussian implements appearance transfer after the rendering, which is not suitable for 3DGS. To maintain the high-speed rendering of 3DGS, we should move all the design components before the rendering starts (the sh coefficients can be cached).
>
> ---
> **W**: "I struggle to understand the motivation for using a triplane to represent local appearance. Any representation that provides a continuous volume, such as triplane, vector and plane components like TensoRF or hash-encoding, could be suitable."
>
> **A**: Triplane is bridge between 2D and 3D, where one can process 3D information in 2D space. Even though 3D representations, such as voxel or octree, can model more complex scenes, they will involve a 3D network to process the 3D information. Compared with 2D networks (2D UNet), 3D networks will introduce more time and space complexities. Therefore, we choose triplane to process and represent the 3D local appearance. However, the decomposition in TensoRF and the structure of hash-encoding are too complex and cannot be processed and predicted by a simple 2D network.
>
> ---
> **W**: "The authors describe a local appearance design intended for physical interactions, such as distinct specular highlights and shadows. However, I have not seen any experiments that demonstrate this, and there is even no ablation experiment without the local feature."
>
> **A**: The Fig. 7 in the Appendix of the main paper contains the results for local appearance modeling. Also, the ablation experiment results without the local feature are provided in the pdf of our rebuttal. Please refer to it.
>
> ---
> **W**: "I am curious to know what the results would be if there were no global and local appearance features."
>
> **A**: Table 1 and Fig. 5 in the main paper give the results for w/o global features. Table 1 and Fig. 1 in our rebuttal pdf present the results for w/o local appearance features and w/o intrinsic feature. Please refer to it.
>
> ---
> **Q**: "I am curious to know about the visual quality, training speed, and rendering speed compared to Scaffold-GS."
>
> **A**: Scaffold-GS can replace the original 3DGS in our method for potentially better performance. Also, we cannot find an in-the-wild dataset that only has the appearance variations between images.
>
> ---
> **Q**: "If the triplane representation is replaced with hash encoding, how would the performance be affected?"
>
> **A**: Hash encoding cannot be simply processed and predicted by a 2D Network.
>
> ---
> **Q**: "It would be beneficial if the authors provided an ablation experiment without the local feature, as it can demonstrate the effectiveness of the local appearance design in modeling distinct specular highlights and shadows."
>
> **A**: Similar question in Weakness. Please refer to the former answers.
>
> ---
> **Q**: "I am interested in knowing the results if there were no global and local appearance features, as it would provide insights into the quality of the decomposition."
>
> **A**: Similar question in Weakness. Please refer to the former answers.
>
> ---

---

> > ### Comment · Reviewer_E9p7 · 2024-08-08
> >
> > Thanks for the author's sincere reply. However, my concern remains partially unresolved.
> > 1. Due to the similarity of the depth regularization and transient objects module with other works, I believe it would be more beneficial to implement a version on Scaffold-GS and then conduct a fair comparison with it, instead of merely stating that "we cannot find an in-the-wild dataset that only exhibits appearance variations between images." That's the correct way to demonstrate novelty and performance.
> > 2. I am still confused as to why triplane cannot be replaced by hash encoding. While the authors argue that hash encoding cannot be easily processed and predicted by a 2D Network, I believe that it's the same with tri-plane.
> > 3. Local appearance is intended to model the specific image's appearance, and I acknowledge its performance. However, the claim regarding "distinct specular highlights and shadows for physical interactions" appears to be overstated.

---

> ### Author Response · Authors · 2024-08-09
>
> Thanks for your prompt and valuable response. We address your remaining concerns below:
>
> ---
> **Concern 1**: "It would be more beneficial to implement a version on Scaffold-GS and then conduct a fair comparison with it."
>
> **Answer**: Following NeRF-W [1], this work focuses on the reconstructing 3D scene from unconstrained photo collections, such as tourism photos over the internet, where the transient objects appear constantly in the existing datasets. However, we agree including Scaffold-GS as additional baseline will be beneficial to better showcase novelty.
>
>
> ***Implementation details***:
>
> Based on the suggestion, we implement an in-the-wild version of Scaffold-GS: (a) For each image, there will be a learnable appearance embedding, and it will be concatenated with original Gaussian features to serve as the input for the color MLP; (b) 2D UNet is leveraged for transient mask prediction and learned in the same way as Wild-GS; \(c\) The appearance embedding should be optimized for each given reference image in the inference stage and this process takes around 5 seconds. Except (b), the entire process is similar to NeRF-W, and we call this version Scaffold-GS-W. We conduct experiments on the six datasets used in the main paper and rebuttal and provide the results below:
>
> Results on Main Paper Datasets:
> | Dataset            | PSNR  | SSIM   | LPIPS  |
> |--------------------|-------|--------|--------|
> | Brandenburg Gate   | 25.55 | 0.9193 | 0.1106 |
> | Sacre Coeur        | 22.77 | 0.8695 | 0.1352 |
> | Trevi Fountain     | 22.08 | 0.7931 | 0.1684 |
>
> Results on Additional Datasets:
>
> | Dataset               | PSNR  | SSIM   | LPIPS  |
> |-----------------------|-------|--------|--------|
> | Palace of Westminster | 23.32 | 0.8611 | 0.1792 |
> | Pantheon Exterior     | 23.48 | 0.8637 | 0.1219 |
> | Buckingham Palace     | 24.90 | 0.8890 | 0.1436 |
>
> Model Efficiency on Single GPU:
>
> | Metric          | Value   |
> |-----------------|---------|
> | Training Time   | 0.15 hrs|
> | Rendering Speed | 192 FPS |
>
>
> ***Observations***:
>
> (a) Scaffold-GS-W significantly outperforms 3DGS-AE (in the rebuttal), especially in SSIM and LPIPS, indicating better reconstruction of local textures and structures.
>
> (b) There is still a big margin on evaluation metrics between Scaffold-GS-W and Wild-GS, even though Wild-GS is based on the original 3DGS.
>
> \(c\) Scaffold-GS-W's inference process is more time-consuming due to the per-image optimization, while Wild-GS offers faster inference by parsing appearance in a single forward pass.
>
> (d) Wild-GS keeps the inference speed of 3DGS, while Scaffold-GS-W shows slightly slower rendering on these datasets compared with 3DGS.
>
> Given the superior reconstruction capabilities of Scaffold-GS over 3DGS, we believe integrating our hierarchical appearance modeling with Scaffold-GS could further enhance Wild-GS's performance. This potential improvement will be discussed in the conclusion and future work section. All the results above will be included in the paper.
>
> [1] NeRF in the Wild: Neural Radiance Fields for Unconstrained Photo Collections
>
> ---
> **Concern 2**: "I am still confused as to why triplane cannot be replaced by hash encoding. While the authors argue that hash encoding cannot be easily processed and predicted by a 2D Network, I believe that it's the same with tri-plane."
>
> **Answer**: Triplane can be processed by 2D networks because each plane (i.e., xy, yz, zx) in triplane is a spatially continuous 2D plane that acts as one "2D projection" of the 3D scene. Using a 2D network (i.e., UNet) to create and process these 2D planes is known to be effective in recent works [1-3]. In contrast, hash encoding [4] is inherently a dictionary/look-up table, which does not have a meaningful 2D spatial structure. Therefore, it cannot be processed by a 2D network that relies on spatial operations.
>
> [1] 3D Neural Field Generation using Triplane Diffusion
>
> [2] Tri-Perspective View for Vision-Based 3D Semantic Occupancy Prediction
>
> [3] RODIN: A Generative Model for Sculpting 3D Digital Avatars Using Diffusion
>
> [4] Instant neural graphics primitives with a multiresolution hash encoding
>
> ---
> **Concern 3**: "Local appearance is intended to model the specific image's appearance, and I acknowledge its performance. However, the claim regarding "distinct specular highlights and shadows for physical interactions" appears to be overstated."
>
> **Answer**: Thanks for your suggestion. We will tone down this statement in the paper.
>
> ---

---

> > ### Comment · Reviewer_TYNp · 2024-08-11
> > **Response to the Reviewer's weaknesses**
> >
> > I thank all reviewers for their diligent efforts in reviewing. However, I publicly condemn expressing such statements as the following:
> >
> > > First, this task is boring because it has already been successfully addressed in NeRF, making it likely that it can also be applied to 3D-GS.
> >
> > - It is inappropriate to label the task as "boring" simply because similar issues have been addressed in NeRF. As researchers, our role is to objectively assess scientific work without letting personal biases, such as deeming a topic "boring", influence our judgment.
> > - The assertion that problems have been resolved by NeRFs is inaccurate. While initial settings like NeRF-in-the-Wild have been explored, there remains substantial potential for improvement in areas such as quality, editability, and training/inference performance. Labeling any problem as "solved" by a particular approach oversimplifies the complexities of scientific research.
> > - The potential applicability of a theory or method, such as Gaussian Splatting's relation to NeRF through the NTK perspective, warrants investigation. It is crucial to explore these avenues thoroughly, regardless of preliminary assumptions about their success.
> >
> > I appreciate the authors for providing additional results with their 3DGS-AE, which contributes valuable insights to the field.

---

> > > ### Comment · Reviewer_E9p7 · 2024-08-11
> > >
> > > Thank you to the authors and reviewers for their sincere efforts in addressing my feedback. Some of my concerns have been resolved, but I still maintain my critical stance.
> > >
> > > 1. Could you clarify why there remains a significant gap in evaluation metrics between Scaffold-GS-W and Wild-GS? Is it due to the inefficacy of using MLP to model appearance?
> > >
> > > 2. I believe the field of novel view synthesis could benefit from fresh insights. There's been a lot of work published recently, all tackling similar problems with the same datasets, which feels somewhat incremental. After reading the paper and especially watching the supplementary video, I found the approach rather uninspiring, particularly for someone who's been in this field for years. Research shouldn't just focus on incremental performance improvements; it should also explore more innovative questions or solutions that contribute meaningfully to the community.
> > >
> > > 3. Please note that I never suggested the problem of novel view synthesis in the wild is "solved." While this method may be somewhat effective, the results aren't surprising and feel incremental, which I believe falls short of the standard for NIPS."

---

> ### Author Response · Authors · 2024-08-12
>
> Thanks for all the comments and discussions from all the reviewers. we would like to continue to address the remaining concerns from Reviewer E9p7:
>
> ---
> **R:** "Could you clarify why there remains a significant gap in evaluation metrics between Scaffold-GS-W and Wild-GS? Is it due to the inefficacy of using MLP to model appearance?"
>
> **A:** No, Wild-GS also utilizes MLP for appearance prediction. We think directly applying the existing appearance modeling methods used in NeRF (i.e., learnable appearance embedding in NeRF-W) to 3DGS without consideration of the explicit and discrete nature of this new representation is suboptimal, which is the reason why there is a gap between Scaffold-GS-W and Wild-GS. Our appearance modeling pipeline is more advanced and effective, and this is one of our contributions in this paper.
>
> ---
> **R:** "I believe the field of novel view synthesis could benefit from fresh insights. There's been a lot of work published recently, all tackling similar problems with the same datasets, which feels somewhat incremental. Research shouldn't just focus on incremental performance improvements."
>
> **A:** First of all, we believe our research is **not incremental**, and we do bring new insights to the field:
> 1) We are the first method that makes real-time rendering from in-the-wild image set possible, and the fast training and real-time rendering performance of Wild-GS will significantly propell the real-world applications of novel view synthesis from unconstrained photo collections.
> 2) Our novel appearance modeling pipeline follows the explicit and discrete nature of 3DGS and will inspire the following works on improving the model performance for this task.
> 3) The extensive experimental analysis and study will serve as a good starting point and expedite the research in this direction.
>
> Secondly, new datasets or new tasks are crucial for advancing a specific research field. However, improving existing tasks using established datasets is equally important for solidifying research progress. For instance, the enhanced performance of the YOLO series [1-4] in real-time object detection has significantly propelled the application of vision models in the real world. Similarly, the evolution of CNN architectures from ResNet [5] to ConvNeXT [6] has laid the groundwork for many vision tasks in recognition, detection, and segmentation. Recently, numerous impressive works [7-10] have adapted 3DGS to existing tasks in novel view synthesis, greatly advancing 3D vision. Therefore, we believe that innovative designs that improve model performance on established tasks are still worth exploring.
>
> [1] You Only Look Once: Unified, Real-Time Object Detection
>
> [2] YOLO9000: Better, Faster, Stronger
>
> [3] YOLOv3: An Incremental Improvement
>
> [4] YOLOv4: Optimal Speed and Accuracy of Object Detection
>
> [5] Deep Residual Learning for Image Recognition
>
> [6] A ConvNet for the 2020s
>
> [7] Human Gaussian Splatting: Real-time Rendering of Animatable Avatars (CVPR 2024)
>
> [8] Text-to-3D using Gaussian Splatting (CVPR 2024)
>
> [9] Dynamic 3D Gaussians: Tracking by Persistent Dynamic View Synthesis (CVPR 2024)
>
> [10] DNGaussian: Optimizing Sparse-View 3D Gaussian Radiance Fields with Global-Local Depth Normalization (CVPR 2024)
>
> ---
> **R:** "After reading the paper and especially watching the supplementary video, I found the approach rather uninspiring, particularly for someone who's been in this field for years."
>
> **A:** As we discussed in the former questions, directly applying existing methods from NeRF to 3DGS is suboptimal, and our Hierarchical Appearance Modeling approach follows the nature of 3DGS and accomplishes more accurate appearance modeling and transfer capabilities than existing methods. Also, the entire Wild-GS pipeline is highly efficient by providing very fast training and similar inference speed with 3DGS, which other existing and concurrent works cannot reach. Therefore, we believe Wild-GS will **inspire the following works on the design of model architecture and appearance modeling pipeline**.
>
> ---
> **R:** "While this method may be somewhat effective, the results aren't surprising and feel incremental, which I believe falls short of the standard for NIPS."
>
> **A:** Wild-GS significantly outperforms existing state-of-the-art models on this task. For instance, compared to CR-NeRF [1], **Wild-GS achieves an approximately 3 PSNR increase while reducing training time by 200 times and increasing rendering speed by 10,000 times**. Even compared with concurrent works, Wild-GS still presents better results. Therefore, from the perspective of experimental results, we still think Wild-GS is **not incremental** and will serve as a good foundation for the following works.
>
> [1] Cross-Ray Neural Radiance Fields for Novel-view Synthesis from Unconstrained Image Collections (ICCV 2023)
>
> ---
> Please let us know if you have any other technical concerns so that we can address them on time.
>
> ---

---

### Official Review · Reviewer_TYNp · 2024-07-02

**Soundness:** 2
**Presentation:** 3
**Contribution:** 3
**Rating:** 6
**Confidence:** 3

**Summary:**

The authors propose a method that adopts recently introduced Gaussian Splatting to work in an in-the-wild setting. The major contribution introduces a decoupling between the global and local changes to the splats. A part of the framework shows how to leverage a given point cloud (from the camera calibration) to condition splats so that they can reproduce local variability in the scene. The experiment results show that the proposed approach improves over the past works, and the introduced components are necessary to obtain them.

**Strengths:**

- The proposed method is novel regarding the Gaussian Splatting applications,
- The qualitative and quantitative results show that the method performs better than the selected baselines.
- Additionally, the ablation study clearly shows that all the components are necessary to obtain the presented results.
- The extraction of features using a point cloud is an interesting novelty that may be used in future research.

**Weaknesses:**

- The model is complex in terms of the number of used components. It uses a pretrained Depth Anything model (for the depth prediction), a 2D UNet that encodes the reference image into a global descriptor, and a 3D UNet that processes the triplane representation. In such a case, how does a method that learns the representations (the global descriptor and triplane representation as local descriptors) in an auto-decoder version (as in NeRF-in-the-Wild) perform?
- As it is the first method (I am not including the preprints here or recently accepted) that applies Gaussian Splatting to the in-the-wild setting, a simpler baseline would strengthen the evaluation. How would a simple learnable per-image latent concatenated with each Gaussian would perform in such a setting?
 - All the chosen baselines have their codebases publicly available. I do not understand why the authors limited their evaluations to 3 scenes only.  In contrast, NeRF-in-the-Wild uses 6 scenes in total from the Phototourism dataset.
- Some parts of the paper need further explanation, notably:
    - Why is the cropping necessary? It would make sense in the case of triplanes that exceed 1024x1024 (for example) pixels in resolution. However, it seems that such a resolution would suffice.
    -  Is the compliment learnable vector $v$ the same for all gaussian outside of AABB?
    - What does "Efficiency" in Table 1. denote exactly?

**Questions:**

I will repeat some of the questions asked already in the weaknesses:
    - Why is the cropping necessary? It would make sense in the case of triplanes that exceed 1024x1024 (for example) pixels in resolution. However, it seems that such a resolution would suffice.
    -  Is the compliment learnable vector `v` the same for all gaussians outside AABB?
    - What exactly does "Efficiency" in Table 1 denote?

I also suggest the authors improve the mathematical notation. For example, the $I_R[M_{I_R} > Th]$ can be decoupled as $I_R \odot \hat{M}_R$ where  $\hat{M}_R = \unicode{x1D7D9}[M_R > \alpha]$. Some symbols can have additional explanations:
- BP in Eq. 6
- $\hat{D}$ in Eq. 10
- $xyz_i$ as $\mathbf{x}_i$

**Limitations:**

The limitations of the paper are clearly stated in the supplementary section. No negative societal impacts are mentioned. However, those do not seem to be of high importance. I suggest the authors mention that next to the limitation section.

---

> ### Author Rebuttal · Authors · 2024-08-07
>
> We sincerely appreciate your thoughtful comments. Below, we address your questions and concerns.
>
> ---
> **W**: "How does a method that learns the representations (the global descriptor and triplane representation as local descriptors) in an auto-decoder version (as in NeRF-in-the-Wild) perform?"
>
> **A**: In the Table 1 of the pdf (our rebuttal), we provide the results for 3DGS-AE (change the encoding (global and local) components to a learnable embedding for each image, work as NeRF-in-the-Wild but keep other components of Wild-GS). As you requested, we also tried to provide a triplane representation (3x1024x1024x32xN) for each image. However, this implementation will cause the out-of-memory issue since the number of training images (N) is too large (around 1000).
>
> ---
> **W**: "How would a simple learnable per-image latent concatenated with each Gaussian would perform in such a setting?"
>
> **A**: Please refer to the former question. We will include this baseline (3DGS-AE) in the main paper.
>
> ---
> **W**: "Why the authors limited their evaluations to 3 scenes only. In contrast, NeRF-in-the-Wild uses 6 scenes in total from the Phototourism dataset. "
>
> **A**: For a fair comparison with Ha-NeRF and CR-NeRF (they only use these 3 scenes), we provide the results of these 3 scenes in the main paper. I guess the reason they limited the experiments to only 3 scenes is because the URLs to download other scenes used NeRF-in-the-Wild became invalid. Considering this, we select 3 extra scenes, which can still be downloaded now, in the Photorism dataset and provide the results (comparing with concurrent work GS-W) in the pdf of our rebuttal. Please refer to it. The new results will be included in the Appendix.
>
> ---
> **Q**: "Why is the cropping necessary? "
>
> **A**: In Table 1 of the main paper, we can notice that after cropping the triplane, the training time can be reduced by around 40\% with even a slight improvement in synthesis performance. The reason is that the majority of Gaussian points only occupy small part of the triplane and cropping the triplane will reduce the complexity for sampling and processing.
>
> ---
> **Q**: "Is the compliment learnable vector the same for all Gaussians outside AABB?"
>
> **A**: Yes, it is the same.
>
> ---
> **Q**: "What exactly does "Efficiency" in Table 1 denote?"
>
> **A**: The efficiency is quantified by training time (number of hours) and inference speed (frame per second) of different models on a single RTX3090 GPU.
>
> ---
> **Q**: "Improve the mathematical notation"
>
> **A**: Thanks so much for your suggestion, and we will change the notation according to your advice.
>
> ---
> **L**: No negative societal impacts are mentioned."
>
> **A**: Thanks for reminding us, and we will mention the societal impacts in the limitation section.
>
> ---

---

> > ### Comment · Reviewer_TYNp · 2024-08-11
> > **Response to the Author's Rebuttal**
> >
> > I appreciate the author's response and the additional results provided. I'm inclined to support the acceptance of the paper for NeurIPS 2024. However, I have reservations about giving a higher score due to certain concerns:
> >
> > - **The tackled problem**: The issue addressed by this paper is well-established within the academic community, with numerous methods already proposed.
> > - **Complexity**: The method involves multiple components to achieve the final results, although I still consider the framework to be novel.
> > - **Datasets**: The datasets utilized have been previously published.
> >
> > Despite these points, rejecting this paper would be a disservice to both the research community and practitioners. Considering that more specialized conferences like ECCV have later deadlines and would require additional work to reach the standards of CVPR, I advocate for the acceptance of this paper at NeurIPS.

---

> > > ### Author Response · Authors · 2024-08-11
> > >
> > > We sincerely appreciate the insightful, objective, and constructive feedback from Reviewer TYNp, and we are grateful for your support in accepting our paper.

---

### Official Review · Reviewer_JabQ · 2024-07-11

**Soundness:** 3
**Presentation:** 3
**Contribution:** 2
**Rating:** 5
**Confidence:** 3

**Summary:**

This paper proposes a new pipeline, which is based on 3D Gaussian Splatting, for in-the-wild rendering. Wild-GS decomposes the appearance into global feature vector, local feature encoded in triplane features and per-Gaussian intrinsic features. Wild-GS achieves best performance on three scenes, while keeping the training and rendering efficiency from 3DGS. However, the novelty is limited since Wild-GS mainly combines existing techniques and replaces NeRF with 3DGS. Besides, the triplane representation may be not suitable if the scenes further scale and become more complex, e.g., self-occlusions.

**Strengths:**

The paper is overall well-written.

Wild-GS achieves better performance than existing methods, while keeping the training and rendering efficiency from 3DGS.

**Weaknesses:**

Limited Novelty. Wild-GS mainly replaces NeRF with 3DGS for the in-the-wild rendering setting and combines existing techniques. Global appearance encoding is estimated in a similar way as Ha-NeRF. Unsupervised visibility mask is also similar to Ha-NeRF. Depth regularization follows FSGS. Combination of triplane features and 3DGS is motivated by previous works, e.g., TriplaneGaussian. Projecting 3D point cloud to generate triplane features is similar to ConvOccNet [1*].

The triplane representation is mainly used in scenes that are not very complex. For example, EG3D works on human faces and objects, while LRM [2*] works on objects. Though the three scenes tested in the paper are buildings, their structure are still relatively simple. If the scenes further scale and become more complex, e.g., self-occlusions, using triplane representation may not be the best choice.

Learned visibility masks should be visualized as Ha-NeRF and CR-NeRF to understand to which extent the model removes transient objects.

[1*] Peng et al. Convolutional Occupancy Networks. ECCV 2020.

[2*] Hong et al. LRM: Large Reconstruction Model for Single Image to 3D. ICLR 2024.

**Questions:**

Is the encoder of the UNet (pretrained ResNet-18) fixed or finetuned?

To get the point cloud with rendered depth, visibility mask is used to remove transient objects. Is there a warm-up process to start using visibility mask since the learned mask may not be good in the beginning of training?

Are the directions of triplanes manually chosen? In Fig.3, the left triplane re-projection nicely corresponds to the front view of the gate.

Is the UNet to process triplane features also pretrained?

What’s the ablation results of not using intrinsic feature?

For appearance transfer, more details need to be discussed. Is the appearance transferred by replacing the $Emb^g$ with the global encoding of the style image?

More details about the video demo should be given. There are two videos in supplementary. Does each video correspond to 2 reference images? What does reference image look like?

**Limitations:**

Limitations are discussed in supplementary. For societal impacts, privacy should be concerned since in-the-wild images usually contain people’s faces.

---

> ### Author Rebuttal · Authors · 2024-08-07
>
> We sincerely appreciate your thoughtful comments. Below, we address your questions and concerns.
>
> ---
> **Q**: "Is the encoder of the UNet (pretrained ResNet-18) fixed or fine-tuned?"
>
> **A**: It is fine-tuned during the training process, which provides better performance than the fixed version.
>
> ---
> **Q**: "Is there a warm-up process to start using visibility mask since the learned mask may not be good in the beginning of training? "
>
> **A**: Yes, as we stated in the paper (line 245), in the initial training stage (3k iterations), we do not use the depth regularization and explicit appearance control strategies, whose functionalities are highly dependent on the mask
>
> ---
> **Q**: "Are the directions of triplanes manually chosen? "
>
> **A**: No, since we project the point cloud to all 6 cube surfaces (all the sides), the direction of the triplane will not affect the results.
>
> ---
> **Q**: "Is the UNet to process triplane features also pretrained? "
>
> **A**: No, the input for this UNet is 6 channels (not traditional RGB.), so we cannot get the pre-trained version.
>
> ---
> **Q**: "What’s the ablation results of not using intrinsic feature? "
>
> **A**: We provide this ablation results in the pdf of our rebuttal, please refer to it. The intrinsic feature is very important for appearance modeling.
>
> ---
> **Q**: "Is the appearance transferred by replacing the 𝐸𝑚𝑏𝑔 with the global encoding of the style image? "
>
> **A**: If the style image is from the dataset (the camera pose is known), we can directly use both global and local encodings for more accurate appearance control. However, if the style image is arbitrary, we should provide an arbitrary camera pose for it to generate the local embeddings, where the global encoding will matter more. We will give more details about this implementation in the Appendix.
>
> ---
> **Q**: "Does each video correspond to 2 reference images? What does reference image look like? "
>
> **A**: Yes. The video is generated by linearly combining 2 reference images (appearance tuning) and changing the camera pose simultaneously. The reference images will be presented in the Appendix for the reader's benefit.
>
> ---
> **L**: "For societal impacts, privacy should be concerned since in-the-wild images usually contain people’s faces."
>
> **A**: That is true. We will recommend users mask the people's faces when leveraging in-the-wild images in the Limitation section.
>
> ---
> **W**: "Wild-GS mainly replaces NeRF with 3DGS for the in-the-wild rendering setting and combines existing techniques."
>
> **A**: Our major contribution in this paper is the establishment of the hierarchical appearance modeling approach following the nature of 3DGS. For each component in this pipeline, one can replace it with different techniques. Through extensive experiments, we finally found that the proposed design is the most effective and efficient, showing better performance than other NeRF-based or 3DGS-based methods.
>
> ---
> **W**: "The triplane representation is mainly used in scenes that are not very complex."
>
> **A**: Even though other explicit representations, such as voxel and octree, can model more complex scenes, the processing networks for 3D inputs are usually very heavy and require more time or space complexity compared with 2D networks. One of the major merits of triplane is that it can move the 3D operation to 2D for better efficiency, which is why we chose triplane. Based on our hierarchical appearance modeling pipeline, one can change the triplane to different representations for explicit appearance control to meet different requirements.
>
> ---
> **W**: "Learned visibility masks should be visualized as Ha-NeRF and CR-NeRF to understand to which extent the model removes transient objects."
>
> **A**: In Fig.2 of the pdf (our rebuttal), we provide the visualizations of the learned masks (threshold by 0.5). One can change to other advanced pre-trained networks (such as DINO v2) for better performance. More mask results will be included in the appendix.
>
> ---

---

> ### Author Response · Authors · 2024-08-12
>
> Dear Reviewer JabQ,
>
> We are truly grateful for the time and effort you have invested in reviewing our paper. We have submitted our responses to your comments, along with a PDF file of the rebuttal. If you have any further questions or need additional clarification, please leave us a comment. We are keen to address any concerns during the discussion period to ensure our manuscript aligns with your expectations.
>
> Thank you once again for your insightful feedback and guidance.
>
> Warm regards,
>
> The Authors

---

> ### Comment · Reviewer_JabQ · 2024-08-13
>
> Thanks for the authors' reply. My questions are answered. About triplane feature, I agree with Reviewer E9p7 that triplane is limited for large scenes. For example, recent SMERF [1*] also points out the limitation of triplane representation in MERF [2*] for large scenes and thus tries to solve it. Take a city scene for example, where the region of interest may contain many buildings. Using triplane features is certainly insufficient because of occlusions. As suggested by Reviewer E9p7, I think using pipelines like Scaffold-GS is an interesting direction. For the computation complexity, since 3D point cloud of 3DGS is a sparse data structure, many methods [3*, 4*] that are specialized to deal with such sparse voxels can be used to improve efficiency.
>
> [1*] Duckworth et al. SMERF: Streamable Memory Efficient Radiance Fields for Real-Time Large-Scene Exploration. SIGGRAPH 2024.
>
> [2*] Reiser et al. MERF: Memory-Efficient Radiance Fields for Real-time View Synthesis in Unbounded Scenes. SIGGRAPH 2023.
>
> [3*] Choy et al. 4D Spatio-Temporal ConvNets: Minkowski Convolutional Neural Networks. CVPR 2019.
>
> [4*] Tang et al. TorchSparse: Efficient Point Cloud Inference Engine. MLSys 2022.

---

> > ### Author Response · Authors · 2024-08-13
> >
> > We are pleased to hear that your questions have been addressed.
> >
> > We agree that replacing the triplane and 2D network with other 3D representations and efficient 3D networks can further enhance the robustness of Wild-GS. Our primary contribution in this paper is the introduction of the appearance modeling pipeline, which involves the decomposition of global, local, and intrinsic features. This approach is particularly suited for 3DGS, and components can be freely replaced with more advanced models based on the specific application scenario of Wild-GS.
> >
> > Thank you once again for your time and effort in reviewing our paper.

---

### Official Review · Reviewer_ZrLe · 2024-07-12

**Soundness:** 3
**Presentation:** 3
**Contribution:** 3
**Rating:** 5
**Confidence:** 4

**Summary:**

**Summary
The paper presents a method called Wild-GS, an adaptation of 3D Gaussian Splatting (3DGS) designed for creating realistic novel views from a collection of unconstrained photographs, such as those taken in varied tourist environments. The method addresses the challenges of dynamic appearances and transient occlusions by employing a hierarchical appearance modeling strategy that decomposes the appearance into global and local components, along with intrinsic material attributes for each 3D Gaussian. Wild-GS introduces an explicit local appearance control using triplane representation, which aligns high-frequency details from reference images to 3D space, and incorporates depth regularization and transient object handling to improve geometric accuracy and rendering quality. Extensive experiments demonstrate Wild-GS's superior performance in rendering efficiency and quality compared to existing techniques, with the promise of publicly available code post-review.

**Strengths:**

**Positive points
1. Wild-GS achieves state-of-the-art rendering performance with significantly improved efficiency in both training and inference times.
2. The method uses triplane representation for explicit local appearance modeling allows for the transfer of high-frequency detailed appearance from reference views to 3D space.

**Weaknesses:**

**Negative points
1. How do you densify and prune the 3D Gaussians, what is the starting iteration and ending iteration and the interval iterations? will the inaccurate mask affect the procedure of the densify and pruning? How does the method perform on the in-the-wild-images? The authors can show more reconstructions in the wild and provide high-resolution images in the main text to make the results of the authors' method more convincing.
2. Will increasing L{m} increasing M{ir}, leading to masking everthing?
3. It would be better to introduce the details of evaluation, for example, NeRF-W, Ha-NeRF and CR-NeRF are test with only half of the images, does the authors uses the same evaluation implementation?
4. From the ablation studies, why does the performance of the baseline method, such as w/o depth in Table.1 also performs well, outperforming existing methods with a large margin?
5. how to ensure the extracted global appearance embedding controls the LF appearance changes?

**Questions:**

Please refer to the weakness section.

**Limitations:**

Yes.

---

> ### Author Rebuttal · Authors · 2024-08-07
>
> We sincerely appreciate your thoughtful comments. Below we address your questions and concerns.
>
> ---
> **Q**: "How do you densify and prune the 3D Gaussians, what is the starting iteration and ending iteration and the interval iterations?"
>
> **A**: For the densification \& pruning and iteration settings, we directly use the default hyperparameters of the original 3DGS. We will recommend the readers to refer to the original implementation of 3DGS.
>
> ---
> **Q**: "Will the inaccurate mask affect the procedure of the densify and pruning?"
>
> **A**: Yes, if the mask cannot locate the transient objects, 3DGS will automatically create lots of Gaussians in these areas to represent their complex and dynamic appearance between different views.
>
> ---
> **Q**: "How does the method perform on the in-the-wild-images?"
>
> **A**: We implemented Wild-GS on three extra datasets consisting of highly "in-the-wild" tourist photos. Please check the quantitative and qualitative results in the pdf of the rebuttal.
>
> ---
> **Q**: "Will increasing L{m} increasing M{ir}, leading to masking everything?"
>
> **A**: Since there is no annotated ground truth for the training images, we need an unsupervised way to learn the visibility mask. The coefficient of the mask loss is a hyperparameter and should not be too large or too small. When it is small, the training process will become unstable, and the regions that are hard to model will be masked, causing geometry errors and even everything being masked. If it is too large, the transient objects cannot be localized.
>
> ---
> **Q**: "NeRF-W, Ha-NeRF and CR-NeRF are test with only half of the images, does the authors uses the same evaluation implementation?"
>
> **A**: NeRF-W optimizes the appearance embedding based on the left half of the image, while recent methods such as Ha-NeRF and CR-NeRF directly encode the appearance features using deep CNN from the whole image (based on their code implementation). Wild-GS follows the same setting as Ha-NeRF and CR-NeRF for a fair comparison and we will provide the results using half of the image in the Appendix.
>
> ---
> **Q**: "Why does the performance of the baseline method, such as w/o depth in Table.1 also performs well, outperforming existing methods with a large margin?"
>
> **A**: The success of Wild-GS mainly originates from the proposed hierarchical appearance modeling. Even though the metric has no significant improvement, the geometry in the rendering results is more accurate. Without depth regularization, sometimes there are some geometry errors in the renderings.
>
> ---
> **Q**: "How to ensure the extracted global appearance embedding controls the LF appearance changes?"
>
> **A**: Wild-GS applies the extracted global appearance embedding to all the 3D Gaussians to capture the global styles or tones for given reference images, ensuring the appearance changes (determined by the global embedding) between Gaussians are insignificant. This is inspired by previous NeRF-based methods (existing baselines NeRF-W and Ha-NeRF), which leverage similar appearance embedding for all the 3D volume points, and their visualization results successfully capture the global low-frequency appearance changes. As shown in Fig. 5 of the paper, with the extracted global embedding, the tones (low-frequency) of the rendering results and references are almost the same and can be transferred to other viewpoints.
>
> ---

---

> > ### Comment · Reviewer_ZrLe · 2024-08-13
> >
> > Thank the authors for answering my questions. I am still confused about the evaluation with only half of the images.  In Table 1 of the manuscript, does the author test with half of the image or use the whole image?

---

> ### Author Response · Authors · 2024-08-12
>
> Dear Reviewer ZrLe,
>
> We sincerely appreciate the time and effort you have dedicated to reviewing our paper. We have submitted our responses to your comments, along with a PDF file of the rebuttal. If you have any additional questions or require further clarification, please let us know. We are eager to address any concerns during the discussion period.
>
> Thank you once again for your valuable feedback and support.
>
> Best wishes,
>
> Authors

---

> ### Author Response · Authors · 2024-08-13
>
> Thanks for your comments.
>
> Following Ha-NeRF and CR-NeRF, we use the whole image to extract the appearance embeddings in Table 1 of the main paper. Ha-NeRF states that "NeRF-W optimizes appearance vectors on the left half of each test image while Ha-NeRF does not", considering NeRF-W "can not hallucinate new appearance
> without optimizing during training." Recent methods (Ha-NeRF and CR-NeRF) directly predict the appearance vector/feature instead of optimizing it for each test image so the whole image can be utilized. Since our method also uses direct appearance encoding, for fair comparison, we follow the same setting. However, for the reader's benefit, we will also attach the evaluation results using only half of the image for testing. Here are the comparison results (average value on three extra datasets used in the rebuttal) of Wild-GS using whole and half of the image:
>
> |       | PSNR    | SSIM   | LPIPS  |
> |-------|:---------:|:--------:|:--------:|
> | Whole | 26.4470 | 0.8788 | 0.1415 |
> | Half  | 26.1326 | 0.8747 | 0.1441 |
>
> There is a slight performance decrease when only half of the image is used. We will also include the results in the manuscript appendix.
>
> If you have any further questions, please let us know. Thanks once again for your efforts.

---

> ### Comment · Reviewer_ZrLe · 2024-08-13
>
> Thanks. But there are still some ambiguities.
> I understand that extracting appearance requires a whole image.
> I only want to know, in Table 1 of the manuscript, in the Brandenburg dataset, do you use half of the image for calculating PSNR SSIM and LPIPS, or the whole image?
> For example,  for a rendered image $\hat{I}$ and the ground truth image $I$. Do you calculate PSNR using $f_{PSNR}(\hat{I}, I$), or do you use $f_{PSNR}(\hat{I}[:, w//2:], I[:,w//2:]$)? Where w is the width of the image, and $f_{PSNR}$ is for calculating PSNR.

---

> > ### Author Response · Authors · 2024-08-13
> >
> > Thanks a lot for your detailed clarification.
> >
> > We use the whole image to calculate all the metrics, and $f_{PSNR}(\hat{I}, I)$ is used to calculate the PSNR.
> >
> > Please let us know if you have any other questions, thanks!

---

> > > ### Comment · Reviewer_ZrLe · 2024-08-13
> > >
> > > Based on my understanding of recent advancements, including NeRF-W, Ha-NeRF, CR-NeRF, and Gaussian-Wild [A], their released code consistently calculates metrics such as PSNR, SSIM, and LPIPS **using only half of the image**.
> > > Moreover, some of the results for existing methods reported by the authors in Table 1 of the manuscript are directly sourced from Table 1 of CR-NeRF, which also employs half-image calculations for these metrics.
> > >
> > > However, the authors have chosen to **calculate metrics on the whole image** in their manuscript and then compare these results with those of other methods that **used only half of the image**. As the authors acknowledged during the rebuttal phase, the metrics computed over the entire image tend to be **higher than** those computed over half of the image.
> > >
> > > This discrepancy leads me to conclude that the comparison presented in Table 1 of the manuscript is not fair. Although the paper still has the potential to outperform existing methods, this issue of unfair comparison must be addressed first.
> > >
> > > [A] Gaussian in the Wild: 3D Gaussian Splatting for Unconstrained Image Collections

---

> ### Author Response · Authors · 2024-08-13
>
> Thanks so much for reminding us of this issue.
>
> The table we provided in the former comment is not **calculated** by the whole and half of the image. We actually misunderstood your question before, and the results reported in that table are generated by using the whole and half of the image to parse the appearance embeddings. Thus, we say, "There is a slight performance decrease when only half of the image is used." because less information is used.
>
> Here are the new results **calculated** by the whole and half of the image:
>
> |       |       |  gate  |        |       |  coeur |        |       | fountain |        |
> |-------|:-----:|:------:|:------:|:-----:|:------:|:------:|:-----:|:--------:|:------:|
> |       |  PSNR |  SSIM  |  LPIPS |  PSNR |  SSIM  |  LPIPS |  PSNR |   SSIM   |  LPIPS |
> | Whole | 29.65 | 0.9333 | 0.0951 | 24.99 | 0.8776 | 0.1270 | 24.45 |  0.8081  | 0.1622 |
> | Half  | 29.92 | 0.9354 | 0.0843 | 25.01 | 0.8797 | 0.1207 | 24.31 |  0.8093  | 0.1593 |
>
> Similar to previous works, these two results do not significantly differ. However, with a scientific and rigorous attitude, we need to greatly thank the reviewer for mentioning this issue. We will replace the metrics in Table 1 of the main paper with these new results on **the right half of the image** ($[w/2, h]$).

---

> > ### Comment · Reviewer_ZrLe · 2024-08-13
> >
> > Thank you for addressing my concern. I will keep my score as borderline accept.

---

> > > ### Author Response · Authors · 2024-08-13
> > >
> > > We are very pleased to be able to address your concern. Thanks a lot for your effort in reviewing our paper.

---

### Author Rebuttal · Authors · 2024-08-07

We thank all reviewers for their valuable comments. We are encouraged that:

- our novelty is recognized (TYNp)
- the superior performance is appreciated (ZrLe, JabQ, TYNp, E9p7)
- the writing clarity is accredited (JabQ, E9p7)

We have tried our best to respond to all the valuable concerns. Please refer to the attached PDF containing figures and tables for the requested experiments.

---

### Decision · Program_Chairs · 2024-09-25

**Decision:**

Accept (poster)

**Comment:**

This paper has received some lengthy discussions. The majority of reviewers (3 out of 4) recommended acceptance (though not highly), while one reviewer recommended rejection with concerns about its novelty.  This AC concurs with the majority of the reviewers that this paper has merits in its real-time performance and high quality output.